# Impact of Psychopathology and Gut Microbiota on Disease Progression in Ulcerative Colitis: A Five-Year Follow-Up Study

**DOI:** 10.3390/microorganisms13061208

**Published:** 2025-05-25

**Authors:** Franco Scaldaferri, Antonio Maria D’Onofrio, Elena Chiera, Adrian Gomez-Nguyen, Gaspare Filippo Ferrajoli, Federica Di Vincenzo, Valentina Petito, Lucrezia Laterza, Daniela Pugliese, Daniele Napolitano, Elisa Schiavoni, Giorgia Spagnolo, Daniele Ferrarese, Lorenza Putignani, Loris Riccardo Lopetuso, Giovanni Cammarota, Fabio Cominelli, Antonio Gasbarrini, Gabriele Sani, Giovanni Camardese

**Affiliations:** 1IBD Unit, Digestive Disease Center (CeMAD), Department of Translational Medicine and Surgery, Fondazione Policlinico Universitario Agostino Gemelli IRCCS, 00168 Rome, Italy; 2CeMAD Translational Research Laboratories, Digestive Disease Center (CeMAD), Department of Medical and Surgical Scences, Fondazione Policlinico Universitario “A. Gemelli”, IRCCS, 44106 Rome, Italy; 3Department of Translational Medicine and Surgery, Catholic University of the Sacred Heart, 00168 Rome, Italy; 4Department of Neuroscience, Section of Psychiatry, Università Cattolica del Sacro Cuore, Largo Francesco Vito 1, 00168 Rome, Italy; 5Università Cattolica del Sacro Cuore, 00168 Rome, Italy; 6Department of Medicine, School of Medicine, Case Western Reserve University, Cleveland, OH 44106, USA; 7Digestive Health Research Institute, School of Medicine, Case Western Reserve University, Cleveland, OH 44106, USA; 8Dipartimento di Salute Mentale, Ospedale di Latina, ASL Latina, 04100 Latina, Italy; 9Unit of Clinical Psychology, Fondazione Policlinico Universitario Agostino Gemelli IRCCS, Università Cattolica del Sacro Cuore, 00168 Rome, Italy; 10Unit of Microbiomics and Unit of Microbiome, Bambino Gesù Children’s Hospital, IRCCS, 00146 Rome, Italy; 11Department of Life Science, Health, and Health Professions, Link Campus University, 00165 Rome, Italy; 12Department of Neuroscience, Section of Psychiatry, Fondazione Policlinico Universitario Agostino Gemelli IRCCS, 00168 Rome, Italy

**Keywords:** ulcerative colitis, gut–brain-axis, gut microbiota, psychopathology, disease activity

## Abstract

Psychological distress and gut dysbiosis play key roles in IBD. This study investigated whether specific psychopathological and gut microbiota features predict adverse outcomes in UC patients. This retrospective cohort study included 35 UC patients recruited in 2019. Baseline assessments involved clinical interviews, psychiatric evaluations, and stool sampling. In 2024, follow-up interviews and medical record reviews assessed disease progression, including biologic therapy failure, hospitalization, surgery, and diagnosis changes. Disease activity was measured via the Mayo score. Psychological testing included MMPI-2, STAI-Y2, GSES, CD-RISC, and TAS-20. Patients with biological therapy failure showed increased levels of Proteobacteria, Fusobacteria, Enterobacteriaceae, and *Trabulsiella*, while Firmicutes were less abundant. UC-related hospitalized patients had lower levels of Rikenellaceae, Ruminococcaceae, *Faecalibacterium*, *Lachnospira*, *Methanobrevibacter*, and *Phascolarctobacterium* compared to non-hospitalized patients. Hospitalized patients scored higher on the Sc clinical scale and the OBS and HEA content scales. *Acidaminococcus* and *Bilophila* were more abundant in patients who underwent surgery. PCA revealed differences between patients with and without biological failure. Logistic regression found that Fusobacteria were negatively correlated with the failure of three or more biologics, while Hy and Pd were positively correlated. Pa and Pt were negatively correlated with multifailure. Obsessiveness, health concerns, somatization, and reduced SCFA-producing bacteria may predict UC-related adverse outcomes.

## 1. Introduction

The gut microbiota–brain axis is a bidirectional communication system between the brain and gut, mediated by a neural pathway [1], an endocrine pathway [2], an immune pathway, and a metabolic pathway [3,4]. This complex network allows the gut microbiota to influence brain function and behavior and, conversely, enables the brain to affect gut physiology and microbial composition [5]. The neural pathway involves the vagus nerve and the enteric nervous system, transmitting signals between the central nervous system (CNS) and the gastrointestinal tract [6]. The endocrine pathway, particularly the hypothalamic–pituitary–adrenal (HPA) axis, regulates stress responses that influence gut barrier integrity and microbial balance [7]. The immune pathway encompasses microbiota-driven modulation of mucosal immunity and cytokine production, which is often dysregulated in inflammatory states [8]. The metabolic pathway includes the microbial production of metabolites—such as short-chain fatty acids (SCFAs) (e.g., butyrate and propionate), tryptophan metabolites, and bile acids—that influence both intestinal homeostasis and CNS signaling [9].

The dysregulation of this axis has been reported in many gastrointestinal diseases, including inflammatory bowel disease (IBD) [10,11]; in fact, gut microbiota dysbiosis plays a critical role in the initiation and persistence of colonic inflammation in IBD, and IBD-associated dysbiosis features have been described in the literature [12]. Zhou et al. highlighted how specific microbiota features were associated with IBD severity [13].

Gut–brain axis dysregulation contributes to psycho-emotional distress in IBD, with anxiety affecting up to one in three patients and depression one in four, rising to one in two and one in three in active disease [14]. Alexithymia, a prevalent trait in IBD, is influenced by disease activity despite its stability [15,16].

Noting the imbalance of the gut microbiota–brain axis in ulcerative colitis (UC), in a previous study by our group we demonstrated that specific gut microbiota signatures are associated with psychopathological profiles in UC [17]. There is growing interest in assessing whether psychopathology and gut microbiota may act as predictors of disease progression or adverse outcomes in IBD, such as surgery or hospitalization. Caenepeel et al. proposed the microbiome-based prediction of response to biologics in IBD [18], while Ananthakrishnan et al. focused specifically on microbiome-based response to anti-integrin biologic therapy [19]. Regarding psychopathology, a systematic review and meta-analysis by Fairbrass et al. [10] evaluated the effects of anxiety and depression on adverse outcomes in IBD, showing that anxiety at baseline was associated with a significantly higher risk of the escalation of therapy-related hospitalization and emergency department attendance, and similar results were obtained with depression. Few studies explore how specific microbiota traits and psychopathological factors (beyond anxiety and depression) influence IBD. This follow-up study aims to assess the psychopathological profile and gut microbiota in UC patients, compare those with adverse outcomes, and identify potential predictors of adverse outcomes.

## 2. Materials and Methods

### 2.1. Study Design

This is a retrospective cohort study that originally recruited 39 UC patients aged at least 20 years [17]. Of the initial 39 patients, 35 were successfully re-contacted and reevaluated in 2024, as some patients had either passed away or were unavailable for follow-up. These 35 patients underwent a clinical interview and physical examination with a gastroenterologist in 2019, which included the collection of personal data, routine demographic information, and the clinical history of their disease. Additionally, a psychiatric interview was conducted to identify any past or current psychiatric disorders, and all participants underwent psychological testing and stool sampling for gut microbiota analysis. Importantly, the psychological assessments and microbiota profiling were conducted in 2019, forming the baseline data for the study.

In 2024, the same cohort was re-contacted by the gastroenterologist for an in-depth clinical interview and an analysis of their medical records to assess the evolution of the pathology over the 5-year follow-up period. Specific outcomes were defined to analyze the progression and evolution of UC from 2019 to 2024, including the failure of biologic therapy, the multifailure of biologic therapy (defined as the failure of at least three biologic agents), UC-related hospitalization, UC-related surgery, and changes in diagnosis (e.g., patients who developed severe perianal disease and had their diagnosis reclassified as Crohn’s disease—CD). These parameters served as adverse outcomes to assess the clinical course of UC over the observation period.

### 2.2. Procedures and Questionnaires

Patients at baseline were assessed using the endoscopic Mayo score and the clinical Mayo score. These two scores together form the full Mayo score, which we used to measure disease activity [20]. Of the 35 patients, *n* = 8 were in remission (score 0–2), *n* = 22 had mild disease activity (score 3–5), *n* = 2 had moderate disease activity (score 6–10), and *n* = 3 had severe disease activity (score > 10).

All patients underwent psychological testing. The following tests were used: (1) the Minnesota Multiphasic Personality Inventory-2 (MMPI-2)—this is the most widely used psychometric test in the world for assessing psychopathology in adults [21]. This test consists of 10 validity subscales, 10 clinical subscales, 31 clinical subscales (divided into the Harris–Lingoes subscale and the Social Introversion subscale), 15 content subscales, 27 subscales related to the components of the content subscales, and 5 additional subscales (PSY-5, Personality Psychopathology Five) [22]. A supplementary table has been provided to summarize the MMPI-2 scales and subscales used in the present study, each accompanied by a brief description to facilitate interpretation (Appendix A). (2) The State–Trait Anxiety Inventory—Form Y (STAI Y1 and Y2)—this is a self-administered test widely used to measure the two dimensions of anxiety, state anxiety (defined as a temporary feeling of tension and apprehension) and trait anxiety (defined as a tendency to be anxious and a general tendency to react anxiously to perceived threats in the environment) [23]. It is characterized by 2 subscales, STAI-Y1 for state anxiety and STAI-Y2 for trait anxiety [24]. For the purposes of the study, only the STAI-Y2 was utilized. (3) The General Self-Efficacy Scale (GSES) measures a person’s optimistic beliefs about their ability to cope with the demands of life. A higher score indicates greater self-efficacy [25]. (4) The Connor–Davidson Resilience Scale (CD-RISC) measures resilience, which is the ability to adapt positively to stress, adversity, and trauma. Higher scores reflect greater resilience [26]. (5) The Toronto Alexithymia Scale-20 (TAS-20) is a self-report measure of alexithymia. It consists of 20 items. It is divided into three subscales——difficulty identifying feelings (7 items), difficulty describing feelings (5 items), and externalizing thinking (8 items). A subject with a total score of 61 or more is considered to be alexithymic [27].

### 2.3. 16S rRNA-Targeted Metagenomics of Fecal Microbiota

According to the manufacturer’s instructions, DNA from stool samples was manually extracted using the QIAmp Fast DNA Stool Mini Kit (Qiagen, Hilden, Germany). The 460-nucleotide (nt) variable region (V3–V4) from the 16S rRNA gene (Primer fw: 16S_F 5ʹ-TCG TCGGCAGCGTCAGATGTGTATAAGAGACAGCCTACGGGNGGCWGC AG)-3ʹ; primer rv: 16S_R 5ʹ (GTCTCGTGGGCTCGGAGATGTGTATAAGAGACAGGACTACHVGGGTATCTAATC C)-3ʹ was amplified by quantitative polymerase chain reaction (qPCR), for each sample, as described in the MiSeq rRNA Amplicon Sequencing protocol (Illumina, San Diego, CA, USA). The first PCR reaction was set up using the following conditions: one step at 95 °C for 3 min, 32 cycles at 95 °C for 30 s, at 55 °C for 30 s, at 72 °C for 30 s, and a final step at 72 °C for 5 min. DNA amplicons were cleaned up by KAPA Pure Beads (Roche Diagnostics, Mannheim, Germany). Indexed libraries were obtained by using Nextera technology (Illumina). The final library was cleaned up using AMPure XP beads and quantified using Quant-iT™ PicoGreen dsDNA Assay Kit (Thermo Fisher Scientific, Waltham, MA, USA). According to the manufacturer’s specifications, the samples were pooled together before the sequencing on an Illumina MiSeqTM platform (Illumina, San Diego, CA, USA) to generate paired-end reads with a base length of 300.

### 2.4. Biocomputational and Statistical Analysis

For the study of patient gut microbiota, the raw metagenomic data generated by the Illumina platform undergo several stages of filtering to obtain sequences that are as accurate as possible, of the expected length, of a quality above Q25, and free of chimeric sequences. Once a dataset of purified sequences is obtained, these are used to generate a data matrix containing information on each identified taxonomic unit (OTU). Relative abundances between OTUs are compared by model-based algorithms using the negative binomial distribution (DESeq2) implemented in R. Tests are filtered for multiple correction tests (FDR or Bonferroni) and only *p*-values < 0.05 after correction are considered significant.

Descriptive statistics were initially computed to summarize the characteristics of the study sample, including measures of central tendency and variability. For continuous variables, mean and standard deviation were used, while categorical variables were presented as frequencies and percentages. To ensure the appropriateness of parametric analyses, we conducted the Shapiro–Wilk test for normality on all total scores. The results indicate that the majority of the scales did not significantly deviate from normality (*p* > 0.05). Moreover, our sample size exceeds 30 participants, which, according to the central limit theorem, allows the sampling distribution of the mean to approximate normality even if the underlying data distribution is not perfectly normal. This further supports the use of parametric statistical methods in our analyses. Regarding the MMPI-2 scales and subscales, in analyzing significant differences in T-scores between groups, we explicitly indicated whenever a score exceeded the clinical cut-off (T ≥ 65) within each group. Although not all scales surpassed this threshold, we consider the observed values to be clinically informative—particularly because the sample does not consist of psychiatric patients. These elevations, even if subclinical, offer valuable insights into psychological functioning and may contribute meaningfully to distinguishing between the groups. Independent t-tests were conducted to compare group means across continuous variables, and chi-square tests were used for categorical variables. To account for multiple comparisons and control the false discovery rate, the Benjamini–Hochberg procedure was applied to all sets of related statistical tests. Principal Component Analysis (PCA) was performed to explore patterns in the gut microbiota composition and MMPI-2 scales across different groups. To further assess the differences between groups identified through PCA, Permutational Multivariate Analysis of Variance (PERMANOVA) was conducted to test for significant group separations. The PERMANOVA results helped confirm the significance of the clustering patterns observed in the PCA. K-means clustering was used to generate unbiased clusters in the PCA. Binary logistic regression analyses were conducted to explore associations between the independent variables (phyla, families, psychometric, and clinical scales) and dependent variables related to the progression of UC, including biological therapy failure and hospitalization. Gender and age were consistently included as covariates in all models. The overall model fit was evaluated using Nagelkerke R^2^. Multiple multinomial regression analyses were also conducted to assess potential interactions between independent variables.

All statistical analyses were performed using SPSS (version 26), and PCA and PERMANOVA were performed in R (version 4.3.2) and RStudio (version 2023.12.0+369).

### 2.5. Ethical Considerations

Ethical approval for this study was obtained from the local ethical committee (ID1886/Prot. N. 0011626/18).

## 3. Results

### 3.1. Demographical, Clinical, Psychometric, and Gut Microbiota Data at Baseline

At baseline, 35 patients with UC diagnosis were enrolled. The inclusion criteria were UC diagnosis that were at least 20 years old, and who were attending the IBD Unit of the CEMAD (Center for Digestive Disease) of “A. Gemelli” IRCCS Hospital in Rome. At baseline, according to the full Mayo score, eight patients were in remission, twenty-two had mild disease activity, two had moderate disease activity, and three had severe disease activity. Twenty-nine were at that moment treated with a biological agent, and six patients underwent experimental therapy. At baseline, nine patients had already experienced failure in reaction to at least one biological agent. The medium age was 40.7 ± 15.2 yrs, and the M:F ratio was 15:20.

### 3.2. Assessment of UC Adverse Outcomes

Twenty-seven patients were under biological therapy at the moment of the follow-up, and twenty-six patients had at least one failure in biological therapy during the five-year observation, while twelve of them were defined as “multifailure”, having not responded to three or more biological agents. Other outcomes that were considered were hospitalization for UC, UC-related surgery, and a change in the diagnosis (patients that were reclassified as CD). The percentage and number of patients in whom these events occurred are shown in Table 1.

### 3.3. Psychopathological and Gut Microbiota Characteristics of Patients with UC Progression

We assessed the differences in gut microbiota composition and psychometric evaluations between patients who experienced a failure of biological therapy (Fail1) and those who did not (nFail1). The results revealed significant differences in gut microbiota composition between the two groups. Specifically, patients with a failure of biological therapy exhibited increased levels of Proteobacteria (t(26.077) = −2.967, *p* = 0.006), Fusobacteria (t(26.382) = −2.134, *p* = 0.042), Enterobacteriaceae (t(26.486) = −2.377, *p* = 0.025), and *Trabulsiella* (t(25) = −2.750, *p* = 0.011), while Firmicutes were found to be less abundant (t(32.279) = 3.450, *p* = 0.036) (Figure 1A). In contrast to the analysis of the microbiota, there were no statistically significant differences in the psychometric assessments between patients with failure of biological therapy and those without.

There were no differences in gut microbiota and psychometric assessments between patients with 3 or more failures of biological therapy (Fail3) and those with no failures or 1–2 failures (nFail3).

Patients hospitalized for complications related to UCs (Hos) presented lower levels of Euryarchaeota (t(28) = 2.200, *p* = 0.036), Rikenellaceae (t(28.008) = 2.710, *p* = 0.011), Ruminococcaceae (t(30.461) = 3.152, *p* = 0.004), *Faecalibacterium* (t(29.770) = 3.117, *p* = 0.004), *Lachnospira* (t(31.129) = 3.013, *p* = 0.005), *Methanobrevibacter* (t(28) = 2.200, *p* = 0.036), *Parabacteroides* (t(32.668) = 2.319, *p* = 0.027), *Collinsella* (t(18.656) = 2.372, *p* = 0.029), and *Phascolarctobacterium* (t(28.096) = 2.723, *p* = 0.011) compared to non-hospitalized patients (nHos) (Figure 1B). Considering the psychometric scales, Hos had higher scores on the Sc clinical scale (t(19.093) = −2.306, *p* = 0.032) and higher scores on the OBS (Obsessiveness) (t(33) = −2.281, *p* = 0.044) and HEA (Health Concerns) (t(33) = −2.324, *p* = 0.026) content scales compared to nHos. Notably, both the OBS and HEA scores exceeded the commonly accepted clinical threshold (T-score ≥ 65), indicating potentially meaningful psychological distress in these domains (Figure 2A).

Considering the patients who underwent surgery for UC (Sur) and non-surgical patients (nSur), the analysis revealed that the two groups differed significantly in gender (χ^2^ = 4.375, *p* = 0.036). To investigate differences in gut microbiota, ANCOVA was conducted, controlling for gender. The results indicated that *Acidaminococcus* (F = 5.749, *p* = 0.023) and *Bilophila* (F = 4.365, *p* = 0.045) were significantly more abundant in the Sur group compared to the nSur group (Figure 1C). No significant differences were observed in the psychometric scales between the two groups.

Patients with other adverse outcomes related to the anatomical extension of the inflammatory process (e.g., severe perianal disease) presented lower levels of *Coprococcus* (t(31.664) = 3.118, *p* = 0.004) compared to patients with no change in their diagnosis (nCh) (Figure 1D). Regarding the psychometric scales, patients in the Ch group had lower scores on the STAI Y2 (t(13.894) = 3.197, *p* = 0.007), higher scores on the clinical scale MF (Masculinity–Femininity) (t(33) = −2.277, *p* = 0.029), and lower scores on the clinical scale Sc (t(33) = 2.089, *p* = 0.044) and the content scale BIZ (Bizarre Mentation) (t(33) = 2.412, *p* = 0.022). Importantly, MF scores in the Ch group exceeded the clinical threshold (T-score ≥ 65), suggesting a potentially significant elevation in this dimension (Figure 2B).

To account for the increased risk of Type I error due to multiple comparisons, *p*-values were corrected using the Benjamini–Hochberg false discovery rate (FDR) method. After correction, all the reported results remained statistically significant, supporting the robustness and reliability of the observed associations.

Several PCA analyses were performed to explore the structure of the data across various dimensions. Initially, a PCA was conducted on the total gut microbiota to identify patterns and potential group separations. This analysis provided insight into the overall composition and variability of the microbiota among the study groups. Subsequently, additional PCA analyses focused on specific taxonomic levels, including phyla, families, and genera. Notably, the PCA and k-means clustering conducted specifically on the phyla revealed distinct clusters and significant differences between patients of Fail1 and those of nFail1 (PERMANOVA: F = 24.2, *p* < 0.001, explained variance = 42.30%) (Figure 3). In contrast, the other PCA analyses—focusing on total microbiota, families, and genera—did not identify any significant differences between the groups.

Parallel to the microbiota analyses, PCA was also performed on the MMPI-2 scales. The first analysis combined both clinical and content scales to identify overarching patterns in psychological assessments. Further analyses were then conducted separately on the clinical scales and the content scales, allowing for a more nuanced understanding of the psychological profiles of the patients. However, these analyses did not reveal significant differences among the groups.

### 3.4. Psychopathological and Gut Microbiota Features as Predictive Factors

Several binary logistic regression analyses were conducted to explore the associations between independent variables and dependent variables related to the progression of UC. The dependent variables included failure of at least one biologic, failure of three or more biologics, hospitalization for reasons related to UC, surgical intervention for UC, and a change in diagnosis. The independent variables encompassed various taxonomic categories, including phyla and families, as well as psychometric scales (specifically STAIY2, CD-RISC, SE, and TAS-20), clinical scales from the MMPI-2, and content scales. Gender and age were consistently included as independent variables in all analyses. A combination of the dependent variables with the independent variables was performed to conduct multiple regression analyses, aiming to identify significant predictive factors for each clinical outcome.

The binary logistic regression analyses identified significant predictors for the failure of three or more biologics related to the phyla of the gut microbiota. Specifically, Fusobacteria was negatively correlated with the failure of three or more biologics, associated with an odds ratio of 0.000 (B = −10,584.860, *p* = 0.028). The overall model fit for the phyla analysis was assessed using the Nagelkerke R^2^, which was found to be 0.415, indicating that the model explained 41.5% of the variance in the failure of three or more biologics, suggesting a moderate fit to the data. Given the presence of extreme odds ratios and potential multicollinearity, we included detailed diagnostic outputs in the Appendix A to allow for a more accurate interpretation of the regression results.

In terms of the clinical scales from the MMPI-2, Hy (Hysteria) and Pd (Psychopathic Deviate) were positively correlated with the failure of three or more biologics, associated with odds ratios of 2.167 (B = 0.773, *p* = 0.044) and 2.639 (B = 0.971, *p* = 0.024). Conversely, Pa (Paranoia) and Pt (Psychastenia) were negatively correlated with the failure of three or more biologics, associated with odds ratios of 0.622 (B = −0.476, *p* = 0.041) and 0.152 (B = −1.886, *p* = 0.021). The overall model fit for the phyla analysis was assessed using the Nagelkerke R^2^, which was found to be 0.790, indicating that the model explained 79% of the variance in the failure of three or more biologics, suggesting a good fit to the data. Importantly, no significant predictors were found for the other outcomes analyzed.

## 4. Discussion

In this study, we aimed to investigate the role of specific gut microbiota compositions and psychometric profiles as factors potentially associated with disease outcomes in UC. By analyzing the associations between microbiological and psychological factors with key clinical endpoints—such as failure of biological therapies, hospitalizations, surgical interventions, and disease progression—we sought to provide new insights into the complex mechanisms underlying UC and to identify potential correlates or indicators of adverse outcomes.

Gut microbiota and psychometric profiles differ by clinical outcomes in ulcerative colitis. Biologic therapy failure was linked to increased Proteobacteria, Fusobacteria, Enterobacteriaceae, and *Trabulsiella*, with reduced Firmicutes but no psychometric differences. Hospitalized patients had lower Ruminococcaceae, *Methanobrevibacter*, *Faecalibacterium*, and *Collinsella*, with elevated obsessive–compulsive and hypochondriacal traits. Surgical patients showed higher *Acidaminococcus* and *Bilophila*, with no psychometric differences. Disease progression was associated with lower *Coprococcus* and distinct psychometric patterns, including lower anxiety but higher specific clinical and content scale scores.

Our findings are consistent with the literature. Lin et al. [28] showed that *Fusobacterium nucleatum* (*F. nucleatum*) may contribute to disease severity in experimental models of DDS-induced colitis by disrupting normal intestinal structure, upregulating inflammatory cytokine expression and leading to intestinal dysbiosis. *F. nucleatum* has also been associated with chronic inflammation and the development of colorectal cancer, including in IBD patients [29,30,31]. For example, *Fusobacterium varium* (*F. varium*) has been found in the colonic mucosa of a high proportion of UC patients [32] and it has been reported that butyric acid, a product of *F. varium* culture supernatants, causes UC-like lesions in mice. An increase in the abundance of Proteobacteria is one of the prominent changes in the microbiota in IBD patients, as described in previous studies [33], and is associated with lower SCFA production and increased colonic inflammation [34]. Barberio et al. 2022 also reported a higher abundance of Proteobacteria in a consistent subset of patients with active UC compared to inactive UC patients and healthy controls, and in another study by Zhou et al. [13], a relative increase in the abundance of Proteobacteria and a relative decrease in Firmicutes were correlated with IBD severity. Particularly, a reduction in the abundance of *Faecalibacterium prausnitzii* (*F. prausnitzii*), a species in the Firmicutes phylum and a producer of SCFAs, was associated with a higher rate of IBD recurrence [35]. With regard to the Firmicutes phylum, the effectors identified as responsible for the anti-inflammatory properties of *F. prausnitzii* are butyrate, an SCFA [36], other products such as shikimic and salicylic acids, molecules involved in inflammation and the maintenance of gut barrier function [37], and a microbial anti-inflammatory molecule (MAM) [38].

Even the reduced abundance of Ruminococcacea, Rikenellaceae, *Lachnospira*, *Phascoarctobacterium*, and *Collinsella*, all SCFA producers [39], has been evaluated in several studies in IBD patients [40,41], suggesting that the reduction in SCFA in the intestinal mucosa could be associated with a worse prognosis in UC patients. A randomized controlled trial by Firoozi et al. [42] showed how butyrate supplementation in patients diagnosed with active ulcerative colitis led to a reduction in inflammation (with a significant reduction in fecal calprotectin), as well as an upregulation of circadian clock genes and an improvement in sleep quality and quality of life, confirming the importance of SCFA in gut–brain communication. The role of *Methanobrevibacter* genus in UC is still unclear—in a study by Scanu et al. [43], UC patients showed an increase in Enterobacteriaceae and a decrease in Ruminococcaceae and *Methanobrevibacter*, although no difference was reported between active and inactive UC patients, while Ghavami et al., 2018, suggest that the decrease in *Methanobrevibacter smithi* may be a microbiome signature of remission [44], which is consistent with our results.

Members of the *Bilophila* genus, such as *Bilophila wadsworthia*, are known as sulfidogenic bacteria that produce hydrogen sulfide from sulfur products in bile or mucus (e.g., taurin) [45]. In this case, our data are consistent with the literature—a study by Pitcher et al. [46] reported a higher number of sulfate-reducing bacteria in patients with active disease compared to those in remission, which also correlated with symptom severity. Furthermore, the decrease in butyrate seen in IBD patients and in UC patients due to gut dysbiosis is reported to decrease the ability to detoxify H2S [47], leading to an alteration of the mucosal barrier [48]. On the other hand, another study shows how the administration of hydrogen sulfide in mouse models leads to a reduction in inflammation and a restoration of microbiota biofilms [49]. Studies on the potential role of the genus *Acidaminococcus* in IBD are limited, although an increase in its abundance in the gut microbiome associated with the intestinal mucosa has been reported in UC patients [50]. This genus, like the previous one, is involved in amino acid metabolism and can ferment glutamate to ammonia, CO_2_, acetate, butyrate, and H2 [51] and seemed to be increased in healthy patients following a pro-inflammatory diet [52].

Our findings on the *Coprococcus* genus are also consistent with the literature; in fact, this genus is causally associated with UC, not CD [53]. Our patients reported a spread of the inflammatory process, leading to the development of perianal fistulas. There are several studies describing the microbial community within CD-associated anorectal fistulas (which appears to be compositionally and functionally unique), although none of them found a change in the abundance of the *Coprococcus* genus [54,55].

With regard to the psychometric tests, patients with higher scores on the Sc (Schizophrenia), OBS (Obsessivity), and HEA (Health Concerns) scales were associated with bad outcomes during the 5-year follow-up period. These results are partially consistent with the previous literature, with Viganò et al. [16], 2018, showing an association with IBD extension and the prevalence of obsessive–compulsive symptoms. The significance of these scales identifies a psychopathological profile characterized by a marked preoccupation with their state of health, often in the absence of serious objectifiable organic symptoms, recurrent thoughts, and, as a likely coping strategy, a tendency towards perfectionism, and may develop compulsions to control the physical symptoms. Between patients that had an extension of inflammation and a change in their diagnosis, there were significantly higher scores on the Masculinity–Femininity scale (Mf), and significantly lower scores on the Schizophrenia scale (Sc) and the Bizarre Mentation scale (BIZ) compared to patients that maintained the UC diagnosis. Furthermore, people who were diagnosed as CD had a lower score in STAI Y2, indicating a lower trait anxiety. The Masculinity–Femininity (Mf) scale is a controversial scale in the MMPI-2—these scales indicate behaviors that deviate from gender stereotypes, which can be a source of difficulty in their relationship with society and their bodies, especially in a state of illness [22].

Our data on trait anxiety are not consistent with the previous literature, which shows that there is a higher prevalence of anxiety symptoms in CD patients compared to UC patients [14]. However, most of the studies included in this systematic review and meta-analysis used measures other than the STAI-Y2 (e.g., the HADS, Hospital Anxiety and Depression Scale) to assess anxiety, many of which were not suitable for distinguishing between trait anxiety and state anxiety. State anxiety has been defined as a transient emotional response involving unpleasant feelings of tension and worrying thoughts. Trait anxiety has been defined as a personality trait that refers to individual differences in the likelihood that a person will experience state anxiety in a stressful situation [56]. A score below the cut-off on the Schizophrenia (Sc) and Bizarre Mentation (BIZ) scales, which correlates with a change in diagnosis, suggests that these patients may tend to think concretely and tend to be rational, conventional, and unimaginative. This suggests a profile similar to alexithymia, characterized by a lack of imagination and introspection.

PCA and k-means clustering, performed to explore the distribution of the data along different dimensions, at the phylum level revealed distinct clusters and significant differences between patients with and without a failure of biological therapy. This suggests that in our cohort of patients there was a specific microbiota signature, considering only the phylum level, that may be associated with a higher risk of biological therapy failure. Several studies in the literature have shown how microbiome-based prediction can help guide the choice of biologic therapy in patients with IBD [13,14]. PCA performed on the MMPI-2 clinical and content scales did not reveal any significant clustering of the data. As suggested by Filipovic et al. [57], this confirms that there is no specific personality type associated with IBD, although some traits seem to be more common than others, as shown by our results.

The binary logistic regression analyses identified significant predictors exclusively for the failure of three or more biologics. Among the gut microbiota, Fusobacteria showed a negative association with this outcome. Regarding the MMPI-2 clinical scales, Hysteria (Hy) and Psychopathic Deviate (Pd) were positively associated with the failure of three or more biologics, while Paranoia (Pa) and Psychasthenia (Pt) exhibited a negative association. Our findings on Fusobacteria seem inconsistent with the previous literature, although most studies focus only on *Fusobacterium nucleatum* and not on other families or genera of this phylum. Given that Fusobacteria are SCFA-producing, we can hypothesize that the reduction in the abundance of the Fusobacteria phylum leads to a decrease in SCFA levels, especially butyrate [58], which is an important regulator of intestinal barrier integrity [59]. This could result in a worsening of intestinal inflammation.

Regarding personality traits in UC patients, it is common for patients with high scores on the Hysteria (Hy) clinical scale to present with health concerns and experience various physical symptoms such as weakness, fatigue, and sleep disturbances [22]. These people tend to be needy and approval-seeking [22]. They avoid conflict and tend to repress their emotions, expressing them through physical discomfort and symptoms [22]. The tendency to somatize makes these patients highly vulnerable and could lead them to perceive the severity of their illness as exaggerated, especially at times of emotional distress, thus creating challenges in the therapeutic alliance between doctor and patient. In addition, the tendency to deny or minimize psychological distress may lead to delayed diagnosis of an underlying anxiety or depressive disorder [60], which, as has been shown in the literature, may worsen the course of IBD [61,62,63,64]. At the same time, a high score on the Psychopathic Deviate scale (Pd) scale indicates a person whose behavior is dominated by impulsivity, a low tendency to conform to social norms, and a distrust of authority figures, including physicians [22]. In some cases, patients may tend to be socially isolated or have difficulty forming lasting relationships, and the poor emotional support they receive may impair their ability to cope with a chronic disease such as UC [65]. High scores on the Paranoia (Pa) and Psychasthenia (Pt) scales emerge as protective factors against the likelihood of failure of three or more biological agents. Patients with high scores on the Paranoia (Pa) scale tend to be suspicious and are extremely cautious in the management and monitoring of their disease [22]. Paranoid patients’ sensitivity to symptoms may facilitate early detection of treatment-related problems and early communication with physicians. At the same time, because of their suspiciousness, they are the patients who, more than others, may benefit from a strong trusting relationship with the physician for an improved outcome in IBD [66]. People who score positive on the Psychasthenia (Pt) scale are perfectionistic, excessively worried, and burdened by intrusive thoughts, but also highly disciplined and detail-oriented [60]. In fact, despite high levels of anxiety, they tend to be very scrupulous and meticulous about managing their disease. These people can be described as ‘ideal patients’. Their tendency to control and their fear of failure drive them to adhere rigorously to treatments, making this personality trait a protective factor, although it undoubtedly affects the patient’s quality of life [67]. It is also important to consider the high levels of stress that this group of patients may experience, which could negatively affect the course of UC [68].

## 5. Limitations

This study has several limitations that warrant careful consideration. First, the overall sample size is modest, and the number of patients in key clinical subgroups—such as those requiring surgery or hospitalization—is particularly small. As a result, subgroup analyses and regression models may be underpowered and should be interpreted with caution. The exploratory nature of these analyses should be emphasized, and conclusions should be framed accordingly. Additionally, although the study spans a five-year follow-up period, gut microbiota was assessed only at baseline. Given the known dynamism of the gut microbiome, the use of a single time point for predictive modeling represents a major limitation. Serial assessments, particularly during disease flares or after critical events such as antibiotic treatment or surgery, could provide important insights into temporal changes in microbiota composition and their clinical relevance. Ongoing follow-up may help to address this gap, but it is important to acknowledge that even in the presence of a generally resilient microbial ecosystem, significant perturbations—such as those induced by prolonged antibiotic use or surgical intervention—are likely to affect microbiota profiles independently of disease activity. These factors should be taken into account when interpreting the findings. While the binary logistic regression analysis identified Fusobacteria as a significant negative predictor of the failure of three or more biologic therapies, the resulting regression coefficient (B = −10,584.860) and odds ratio (OR = 0.000) suggest the presence of quasi-complete separation in the data. This statistical artifact typically arises when a predictor variable perfectly or nearly perfectly predicts the outcome, often due to a small sample size or highly imbalanced group distributions. Although the *p*-value was below the significance threshold, the extreme nature of the coefficient and OR indicates instability in the model estimates and limits the interpretability and generalizability of this finding. These results should therefore be considered exploratory, and future studies with larger sample sizes and more balanced group distributions are needed to confirm the association between Fusobacteria and biologic treatment failure in patients with ulcerative colitis.

## 6. Conclusions and Future Perspectives

Our study is a cohort, retrospective, interventional study. The results show the presence of relevant psychopathological distress in patients with ulcerative colitis, although there is no single personality type common to patients with this pathology, as also highlighted in the literature. Psychopathological traits (especially related to obsessiveness, health concerns, somatization) and some microbiota features (a lower abundance of SCFA producers bacteria) at baseline can act as predictive factors to assess UC-related adverse outcomes.

As a future perspective, serial assessments of the gut microbiota and a psychological picture of the patients in relation to specific events in the course of the disease could provide a more personalized and comprehensive evaluation. Personality, being a construct that is stable over time and relatively easy to assess, could be a useful risk assessment tool in IBD, moving towards a “psychogastronterological approach” in these diseases.

## Figures and Tables

**Figure 1 microorganisms-13-01208-f001:**
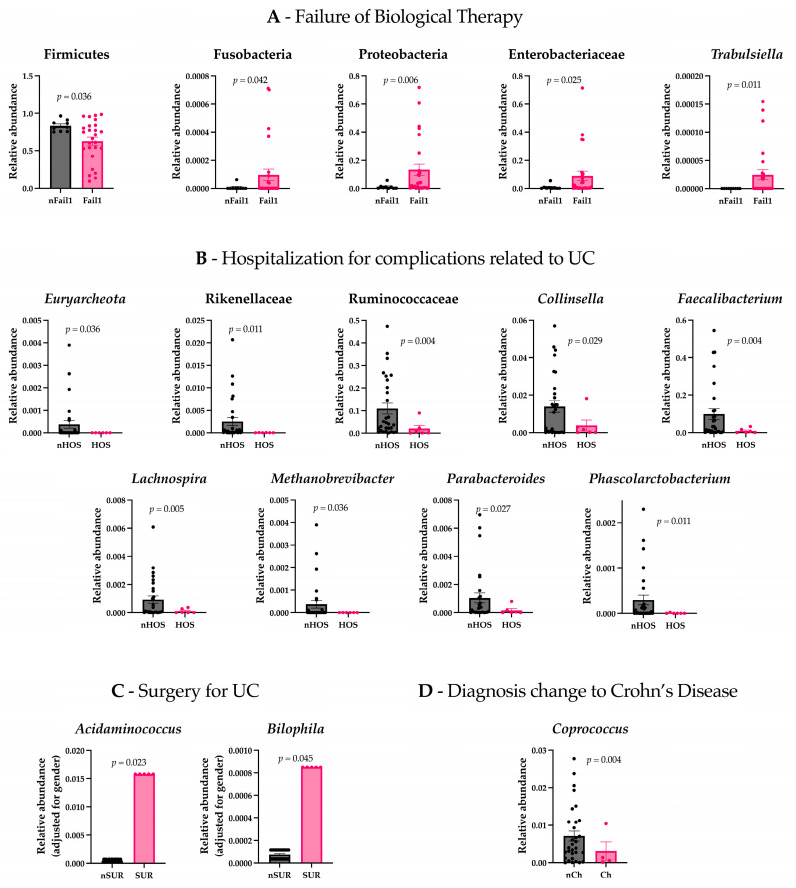
Gut microbiota differences between patient groups. (**A**) Patients who experienced a failure of biological therapy (Fail1) showed significantly increased levels of Proteobacteria, Fusobacteria, Enterobacteriaceae, and *Trabulsiella*, and decreased Firmicutes compared to patients who did not experience failure (nFail1). (**B**) Patients hospitalized for complications related to ulcerative colitis (Hos) exhibited lower levels of *Euryarchaeota*, Rikenellaceae, Ruminococcaceae, *Faecalibacterium*, *Lachnospira*, *Methanobrevibacter*, *Parabacteroides*, *Phascolarctobacterium*, and *Collinsella* compared to non-hospitalized patients (nHos). (**C**) Patients who underwent surgery for UC (Sur) had higher levels of *Acidaminococcus* and *Bilophila* compared to non-surgical patients (nSur). (**D**) Patients with a diagnosis change to Crohn’s disease (Ch) had lower levels of *Coprococcus* compared to those with no diagnosis change (nCh).

**Figure 2 microorganisms-13-01208-f002:**
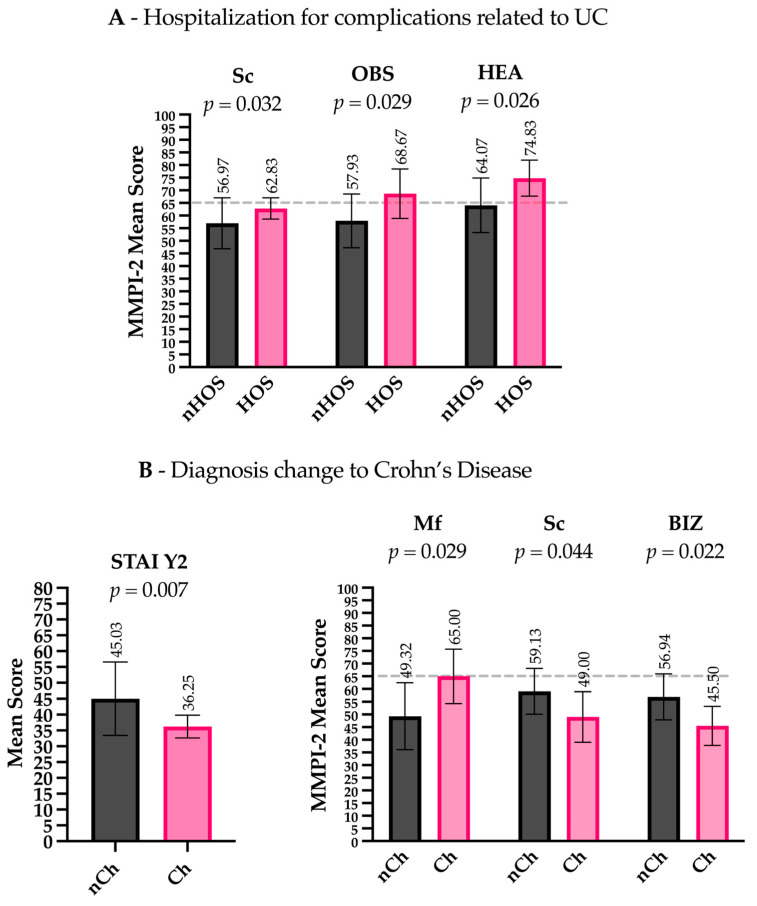
Psychometric scale differences between patient groups. (**A**) Hospitalized patients (Hos) had higher scores on the **Sc** (Schizophrenia) clinical scale, as well as on the OBS (Obsessiveness) and HEA (Health Concerns) content scales, compared to non-hospitalized patients (nHos). (**B**) Patients with a diagnosis change to CD (Ch) scored lower on the STAI Y2 (trait anxiety), higher on the Mf (Masculinity–Femininity) clinical scale, and lower on the Sc (Schizophrenia) clinical scale and the BIZ (Bizarre Mentation) content scale compared to those without a diagnosis change (nCh). Gray dashed lines indicate the clinical threshold (T-score ≥ 65) above which scores are generally considered clinically significant.

**Figure 3 microorganisms-13-01208-f003:**
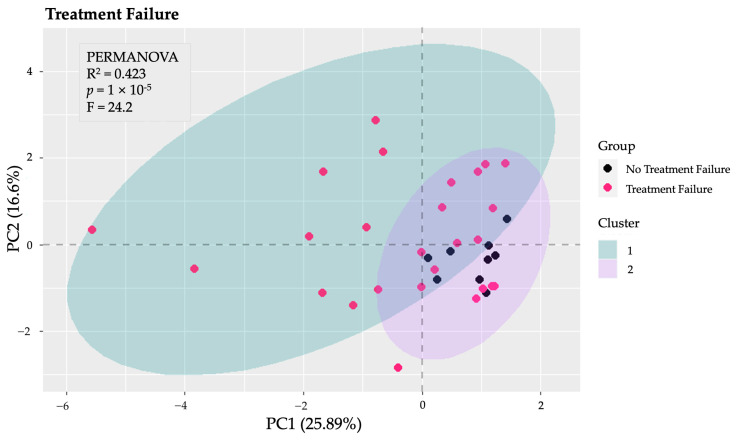
Principal Component Analysis (PCA) of gut microbiota at the phylum level. The PCA conducted on the phyla of the gut microbiota revealed significant differences between patients who experienced the failure of biological therapy (Fail1) and those who did not (nFail1), as indicated by PERMANOVA (*p* < 0.001), with an explained variance of 42.3%.

**Table 1 microorganisms-13-01208-t001:** Failure of biological therapy, therapy multifailure, UC-related surgery, UC-related hospitalization, and reclassification of diagnosis to CD were used as outcomes to assess the progression of the disease during the 5-year follow-up.

**Therapy failure (*n*, %)**	No (9, 25.7%)
	Yes (26, 74.3%)
**Therapy multifailure (>3) (n, %)**	No (23, 65.8%)
	Yes (12, 34.2%)
**Surgery (UC-related) (n, %)**	No (30, 85.7%)
	Yes (5, 14.3%)
**Hospitalization (UC-related) (n, %)**	No (29, 82.9%)
	Yes (6, 17.1%)
**Variation in diagnosis (CD) (n, %)**	No (31, 88.6%)
	Yes (4, 11.4%)

## Data Availability

The datasets generated and/or analyzed during the current study are not publicly available due to ethical restrictions and the sensitive nature of the clinical and psychometric data. However, deidentified data may be made available by the corresponding author upon reasonable request and with approval from the local ethics committee.

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
