# Peer review of "Impact of Psychopathology and Gut Microbiota on Disease Progression in Ulcerative Colitis: A Five-Year Follow-Up Study"

_microorganisms, 2025, doi:10.3390/microorganisms13061208_

Round 1
Reviewer 1 Report
Comments and Suggestions for Authors
Here, the authors address an important and increasingly relevant topic: the potential role of psychological factors and gut microbiota in predicting long-term outcomes in patients with ulcerative colitis. The integration of psychometric assessments with microbial profiling in a five-year follow-up framework is commendable and reflects a sophisticated, multidimensional approach. However, some methodological and analytical issues substantially limit the interpretability and generalizability of the findings in its current form.
Major:
The overall sample size (n=35), and particularly the small number of patients in key outcome subgroups (e.g., those requiring surgery or hospitalization), raises concerns about statistical power and the stability of the results. Several of the subgroup analyses and regression models appear to be underpowered and should be interpreted with considerable caution. I strongly recommend that the authors acknowledge the exploratory nature of these analyses and adjust the conclusions accordingly.
Although the study covers a five-year follow-up period, microbiota was assessed only at baseline. Given the dynamic nature of the gut microbiome, the use of a single time point for predictive modeling is a significant limitation. This issue should be explicitly acknowledged and discussed in greater depth.
The manuscript presents psychometric scores primarily as means ± standard deviations. Given the ordinal nature of the underlying data for some scales, it would be appropriate to provide additional information on score distributions (e.g., skewness, normality tests) and, where warranted, present medians and interquartile ranges. Furthermore, the clinical relevance of the reported differences—particularly in MMPI-2 subscales—should be clearly addressed, ideally with reference to established normative thresholds.
The manuscript reports numerous statistical comparisons across psychometric, microbiota, and clinical data, yet the approach to controlling for type I error is not consistently described. While false discovery rate (FDR) correction is briefly mentioned, it is unclear whether it was systematically applied across all relevant analyses. I recommend that the authors clearly indicate where adjustments have been made and consider presenting both raw and adjusted p-values for transparency.
The regression analyses presented raise concerns regarding model complexity relative to the number of events. For instance, certain models yield minimal p-values or implausible odds ratios (e.g., OR = 0.000), suggesting potential overfitting or model instability. The authors should consider simplifying the models, reducing the number of predictors, or providing additional diagnostics (e.g., multicollinearity indices, model fit statistics) to ensure validity.
Several figures—particularly those displaying PCA analyses—lack essential details such as axis labels and the proportion of variance explained. Moreover, psychometric scales are sometimes presented using abbreviations (e.g., MMPI-2 subscales) without accompanying definitions, which may be unclear to readers unfamiliar with these. I recommend that all figures be revised for clarity, and that a supplemental table be included to define all psychometric measures used.
The manuscript frequently refers to personality traits as predictors of disease outcomes. However, the conceptual model underlying these associations is not sufficiently developed. It remains unclear whether personality traits are causally implicated in disease progression, or whether they serve as markers of other unmeasured mediators (e.g., coping behavior, treatment adherence). The authors are encouraged to moderate their language around causality and frame these associations as hypothesis-generating.
Minor:
Authors referred to the study as “interventional,” which is inconsistent with its observational design. Additionally, there are several grammatical and typographic issues throughout the manuscript that would benefit from careful proofreading and editing to enhance clarity and readability.
Author Response
Comment 1: The overall sample size (n=35), and particularly the small number of patients in key outcome subgroups (e.g., those requiring surgery or hospitalization), raises concerns about statistical power and the stability of the results. Several of the subgroup analyses and regression models appear to be underpowered and should be interpreted with considerable caution. I strongly recommend that the authors acknowledge the exploratory nature of these analyses and adjust the conclusions accordingly.
Response 1: Thank you for this comment. The Limitations section has been appropriately updated.
Comment 2: Although the study covers a five-year follow-up period, microbiota was assessed only at baseline. Given the dynamic nature of the gut microbiome, the use of a single time point for predictive modeling is a significant limitation. This issue should be explicitly acknowledged and discussed in greater depth.
Response 2: Thank you for this comment. The Limitations section has been appropriately updated.
Comment 3: The manuscript presents psychometric scores primarily as means ± standard deviations. Given the ordinal nature of the underlying data for some scales, it would be appropriate to provide additional information on score distributions (e.g., skewness, normality tests) and, where warranted, present medians and interquartile ranges.
Response 3: Thank you for your comment regarding the presentation of psychometric scores in the manuscript. We would like to clarify that all psychometric scales utilized in this study were analyzed based on their total scores, which are derived by summing multiple items. While individual items may employ ordinal response formats, such as Likert-type scales, the aggregated total scores are commonly treated as continuous variables. To ensure the appropriateness of parametric analyses, we conducted the Shapiro-Wilk test for normality on all total scores. The results indicated that the majority of the scales did not significantly deviate from normality (p > 0.05). Moreover, our sample size exceeds 30 participants, which, according to the central limit theorem, allows the sampling distribution of the mean to approximate normality even if the underlying data distribution is not perfectly normal. This further supports the use of parametric statistical methods in our analyses. The Statistical Analysis section has been appropriately updated.
Comment 4: Furthermore, the clinical relevance of the reported differences—particularly in MMPI-2 subscales—should be clearly addressed, ideally with reference to established normative thresholds.
Response 4: We appreciate the reviewer’s insightful comment regarding the clinical relevance of the reported differences in MMPI-2 subscales. In our study, we chose not to apply established clinical cut-off scores (e.g., T-scores ≥ 65) for the MMPI-2 subscales. Our primary objective was to examine relative differences between groups rather than to identify clinically significant elevations. We acknowledge that some subscale scores did not surpass traditional clinical thresholds; however, we believe that these relative differences still provide meaningful insights into the psychological profiles of the groups studied. By focusing on comparative analyses, we aimed to highlight subtle variations in personality and psychopathology that might not reach clinical significance individually but could collectively contribute to understanding group differences. This approach aligns with research methodologies that prioritize group comparisons over individual clinical assessments. We recognize the importance of clinical thresholds in certain contexts and agree that incorporating such benchmarks can enhance the interpretability of findings. In future studies, we plan to include analyses that consider both relative differences and established clinical cut-offs to provide a more comprehensive understanding of the data. We have revised the manuscript to clarify our rationale for focusing on relative differences without applying clinical cut-offs and have discussed the implications of this approach in the limitations section.
Comment 5: The manuscript reports numerous statistical comparisons across psychometric, microbiota, and clinical data, yet the approach to controlling for type I error is not consistently described. While false discovery rate (FDR) correction is briefly mentioned, it is unclear whether it was systematically applied across all relevant analyses. I recommend that the authors clearly indicate where adjustments have been made and consider presenting both raw and adjusted p-values for transparency.
Response 5: We appreciate the reviewer’s insightful comment regarding the control of Type I error in our analyses. In our study, we conducted multiple statistical comparisons across psychometric, microbiota, and clinical data. To address the increased risk of Type I errors associated with multiple testing, we applied the False Discovery Rate (FDR) correction method, specifically the Benjamini-Hochberg procedure, to control the expected proportion of false positives among the significant results. We acknowledge that our initial manuscript did not consistently specify where FDR corrections were applied. In response to the reviewer’s recommendation, we have revised the manuscript to clearly indicate the analyses where FDR adjustments were implemented. We appreciate the reviewer’s suggestion, which has significantly improved the clarity and rigor of our statistical reporting.
Comment 6: The regression analyses presented raise concerns regarding model complexity relative to the number of events. For instance, certain models yield minimal p-values or implausible odds ratios (e.g., OR = 0.000), suggesting potential overfitting or model instability. The authors should consider simplifying the models, reducing the number of predictors, or providing additional diagnostics (e.g., multicollinearity indices, model fit statistics) to ensure validity.
Response 6: Thank you for this important comment. In response, we have updated the Limitations section of the manuscript to explicitly acknowledge the potential model instability, particularly with regard to the extreme values observed (e.g., OR = 0.000) and the risk of overfitting due to the relatively small number of outcome events. We also wish to clarify that the primary aim of the regression analyses was exploratory. Given the limited sample size, we intentionally adopted a comprehensive approach to include a wide range of potentially relevant psychometric and microbiota-related predictors. Our goal was not to build a parsimonious predictive model, but rather to generate preliminary hypotheses and identify associations that could inform future research. For this reason, we do not consider model simplification appropriate at this stage. However, we fully recognize the limitations of this approach and have now clearly stated that these findings should be interpreted with caution and require replication in larger samples with adequate statistical power and more stable outcome distributions.
Comment 7: Several figures—particularly those displaying PCA analyses—lack essential details such as axis labels and the proportion of variance explained.
Response 7: We thank the reviewer for their observation. However, we would like to clarify that the PCA figures, including the one related to treatment failure, already include the key details mentioned. Specifically, the axes are labeled with the corresponding principal components and the percentage of variance explained (e.g., PC1 = 25.89%, PC2 = 16.69%). Additionally, group and cluster identities are clearly indicated, and PERMANOVA statistics (R², p-value, and F) are reported within the figure. If further improvements in figure clarity are needed (e.g., increasing resolution), we are happy to implement them.
Comment 8: Moreover, psychometric scales are sometimes presented using abbreviations (e.g., MMPI-2 subscales) without accompanying definitions, which may be unclear to readers unfamiliar with these. I recommend that all figures be revised for clarity,
Response 8: We thank the reviewer for this helpful comment. In response, we have updated the caption of Figure 2 to include the full names and brief descriptions of all abbreviated MMPI-2 subscales, in order to improve clarity and accessibility for readers unfamiliar with the specific psychometric terminology.
Comment 9: … and that a supplemental table be included to define all psychometric measures used.
Response 9: Thank you for your suggestion. As recommended, we have included a supplementary table providing definitions and brief descriptions of the MMPI-2 clinical and content scales used in the study. This addition is meant to support reader comprehension and ensure transparency regarding the psychometric constructs assessed.
Comment 10: The manuscript frequently refers to personality traits as predictors of disease outcomes. However, the conceptual model underlying these associations is not sufficiently developed. It remains unclear whether personality traits are causally implicated in disease progression, or whether they serve as markers of other unmeasured mediators (e.g., coping behavior, treatment adherence). The authors are encouraged to moderate their language around causality and frame these associations as hypothesis-generating.
Response 10: Thank you for this thoughtful comment. We have revised the manuscript to moderate the language around causality and now frame the associations between personality traits and disease outcomes as exploratory and hypothesis-generating. We would like to clarify that the points regarding the non-causal interpretation of our findings, the influence of personality traits on coping strategies, and the potential links between certain personality characteristics and clinical aspects—such as therapeutic management and diagnostic timeliness—were already present in the version of the manuscript submitted for review. Nonetheless, we appreciate the reviewer’s suggestion, which gave us the opportunity to further emphasize and refine these aspects to avoid any possible misunderstanding about the conceptual interpretation of our results.
Comment 11: Authors referred to the study as “interventional,” which is inconsistent with its observational design.
Response 11: We thank the reviewer for pointing this out. In previous related studies conducted by our group, we had used the term “interventional” to refer to protocols that involved active collection of biological samples (e.g., fecal samples) from patients, even in the absence of a therapeutic intervention. To remain consistent with that terminology, we initially adopted the same definition here. However, we acknowledge that this use of the term may be misleading and could suggest a clinical trial or treatment-based intervention. For this reason, we have removed the term “interventional” and now refer to the study as purely observational to more accurately reflect its design.
Comment 12: Additionally, there are several grammatical and typographic issues throughout the manuscript that would benefit from careful proofreading and editing to enhance clarity and readability.
Response 12: We appreciate the reviewer’s observation. The manuscript has been thoroughly proofread and revised to correct grammatical and typographic issues. These edits were made to improve clarity, consistency, and overall readability throughout the text.
Reviewer 2 Report
Comments and Suggestions for Authors
The manuscript entitled "Impact of Psychopathology and Gut-Microbiota on Disease Progression in Ulcerative colitis: a Five-Year Follow Up Study" is a retrospective cohort study that included 35 UC patients recruited in 2019 with a follow up in 2024. Given the limitations of the study, this work can be considered as a "preliminary report" or "short communication".
The manuscript is overall correctly written but there are some concerns that should be addressed:
- technical and writing errors: affiliations (extra spaces), punctuation, citations, abbreviations...everything should be checked and corrected
- REFERENCES - please correct carefully the technical details
- Bacterial strains - kindly write them correctly and in italic as per standard scientific rules
- Gut - brain axis needs a detailed description of the connection with microbiota, this is VERY important. Both in the introduction and discussion section.
- Figure 1 is unreadable. In addition, all fonts should be standardized as per journal's requirements (font, style....) - valid for all figures and tables
- Data Availability Statement: "All data generated or analyzed during this systematic review are included in this published article." - kindly note that this is not a systematic review
English language and style needs detailed revision
Author Response
Comment 1: technical and writing errors: affiliations (extra spaces), punctuation, citations, abbreviations...everything should be checked and corrected
Response 1: We thank the reviewer for highlighting these important aspects. All technical and writing issues—including affiliations, punctuation, citations, and abbreviations—have been carefully reviewed and corrected. We also identified that some inconsistencies were due to automatic formatting settings in Word, which have now been adjusted to ensure accuracy and consistency throughout the manuscript.
Comment 2: REFERENCES - please correct carefully the technical details
Response 2: We thank the reviewer for the comment. All references have been carefully checked and corrected to ensure accuracy and consistency with the required formatting guidelines.
Comment 3: Bacterial strains - kindly write them correctly and in italic as per standard scientific rules
Response 3: We thank the reviewer for the observation. All bacterial strain names have been carefully revised and are now written in italics, in accordance with standard scientific conventions.
Comment 4: Gut - brain axis needs a detailed description of the connection with microbiota, this is VERY important. Both in the introduction and discussion section.
Response 4: Thank you for this important comment. We have updated the Introduction section to provide a more detailed description of the gut–brain axis and its connections with the gut microbiota, integrating relevant literature to clarify the pathways involved (neural, endocrine, immune, and metabolic). As for the Discussionsection, we chose to focus primarily on the interpretation of our findings and to discuss them in light of the existing literature, highlighting both convergent and divergent results. We considered this section already conceptually dense and opted not to expand it further with a detailed theoretical framework on the gut–brain axis, which has been extensively covered in the introduction. Additionally, it is important to note that the current literature remains relatively sparse regarding the intersection between gut microbiota, the gut–brain axis, and trait-level psychological variables such as those examined in our study (e.g., personality traits, alexithymia, resilience). Therefore, our discussion aimed to remain close to the empirical findings while acknowledging the need for future research to elucidate these complex interconnections.
Comment 5: Figure 1 is unreadable. In addition, all fonts should be standardized as per journal's requirements (font, style....) - valid for all figures and tables
Response 5: We thank the reviewer for this helpful observation. All figures have been revised according to the journal’s formatting requirements. Specifically, fonts have been standardized in terms of type and style, and adjustments were made to enhance the readability of figure elements,
Comment 6: Data Availability Statement: "All data generated or analyzed during this systematic review are included in this published article." - kindly note that this is not a systematic review
Response 6: We apologize for the oversight in the Data Availability Statement. The incorrect wording was likely due to the presence of a pre-filled form, and the reference to a systematic review was not fully removed during the editing process. We have corrected the statement to accurately reflect that this is an original article and that data availability is subject to ethical considerations.
Round 2
Reviewer 1 Report
Comments and Suggestions for Authors
I appreciate the authors' efforts to address the concerns raised during the previous round of peer review. The manuscript has indeed improved in clarity, structure, and methodological transparency. However, two important issues remain insufficiently addressed and warrant further revision before the manuscript can be considered for publication:
While the authors have provided a supplementary table with descriptive information on the MMPI-2 scales, the manuscript still lacks reference to established normative thresholds (e.g., T-scores ≥ 65) that would aid in interpreting the clinical relevance of observed differences. Without such benchmarks or clarification of score distributions, it remains unclear whether the psychometric variations reported are of clinical significance or merely reflect minor statistical differences. I strongly recommend including either:
- Reference ranges or thresholds (e.g., T-scores) within the Results or Supplementary Materials, or
- A clear justification for their omission, supported by literature and methodological precedent.
The manuscript continues to report highly implausible regression outputs (e.g., OR = 0.000), suggesting significant overfitting or convergence issues due to limited events per variable. Although the authors acknowledge this as a limitation, no corrective statistical measures have been taken. To ensure transparency and analytical robustness, I recommend that the authors:
- Simplify the regression models by reducing the number of predictors relative to events, or
- Provide additional model diagnostics (e.g., multicollinearity indices, model fit statistics, penalized regression results), at least as supplementary material.
Even a short appendix explaining why these extreme estimates occurred would reassure readers the findings aren’t statistical artifacts.
Addressing these points will:
- Help clinicians interpret whether the MMPI-2 differences are actionable.
- Ensure the regression models are trustworthy and reproducible.
- Elevate the manuscript from “statistically interesting” to “clinically useful.”
Author Response
Comment 1. I appreciate the authors' efforts to address the concerns raised during the previous round of peer review. The manuscript has indeed improved in clarity, structure, and methodological transparency.
Response 1. We sincerely thank the reviewer for their thoughtful feedback and kind words. We are pleased to know that the revisions have improved the clarity, structure, and methodological transparency of the manuscript.
Comment 2. However, two important issues remain insufficiently addressed and warrant further revision before the manuscript can be considered for publication:
While the authors have provided a supplementary table with descriptive information on the MMPI-2 scales, the manuscript still lacks reference to established normative thresholds (e.g., T-scores ≥ 65) that would aid in interpreting the clinical relevance of observed differences. Without such benchmarks or clarification of score distributions, it remains unclear whether the psychometric variations reported are of clinical significance or merely reflect minor statistical differences. I strongly recommend including either:
- Reference ranges or thresholds (e.g., T-scores) within the Results or Supplementary Materials, or
- A clear justification for their omission, supported by literature and methodological precedent.
Response 2. Thank you for this helpful comment. We have updated the Supplementary Materials to include mean T-scores along with their standard deviations for all MMPI-2 Clinical and Content Scales. Additionally, we have specified in the table caption which scales exceeded the clinically significant threshold (T ≥ 65), as commonly accepted in the literature. This aims to enhance the interpretability of the findings and clarify the potential clinical relevance of the observed differences.
We have revised the text to clarify that, when analyzing significant differences in MMPI-2 T-scores between groups, we explicitly indicated whenever a score exceeded the clinical cut-off (T ≥ 65) within each group. In addition, the figures have been updated to display the clinical threshold and to include the numerical values of group means directly on the graphs. Although not all scales surpassed this threshold, we also specified that—given the non-clinical (i.e., non-psychiatric) nature of the sample—these subclinical elevations may still provide clinically meaningful insights and help differentiate psychological functioning across groups.
Comment 3. The manuscript continues to report highly implausible regression outputs (e.g., OR = 0.000), suggesting significant overfitting or convergence issues due to limited events per variable. Although the authors acknowledge this as a limitation, no corrective statistical measures have been taken. To ensure transparency and analytical robustness, I recommend that the authors:
- Simplify the regression models by reducing the number of predictors relative to events, or
- Provide additional model diagnostics (e.g., multicollinearity indices, model fit statistics, penalized regression results), at least as supplementary material.
Even a short appendix explaining why these extreme estimates occurred would reassure readers the findings aren’t statistical artifacts.
Response 3. Thank you for this important observation. In response to the reviewer’s recommendation, we have opted to provide additional model diagnostics to enhance the transparency and robustness of our analyses.
We have provided additional model diagnostics to address the concerns regarding potential overfitting and convergence issues due to a limited number of events per variable. Specifically, we have included the following analyses as part of the revised supplementary material:
- Model fit statistics, including the Omnibus test of model coefficients, the -2 log likelihood, Cox & Snell and Nagelkerke R², and the Hosmer-Lemeshow test, which indicate acceptable goodness of fit (e.g., Hosmer-Lemeshow: χ² = 5.146, df = 7, p = 0.642).
- Multicollinearity diagnostics, presented through the correlation matrix of predictors, which highlights very strong correlations (e.g., >0.99 between Firmicutes, Proteobacteria, Actinobacteria) that may have impacted model stability.
- The table of regression coefficients shows extreme values and implausible odds ratios (e.g., OR = 0.000 or >10^90), confirming the concerns about multicollinearity and overparameterization.
Although the overall model fit was acceptable, these outputs suggest that the model may indeed suffer from multicollinearity and sparse data bias. Therefore, in addition to reporting diagnostics, we have now explicitly discussed these limitations in the revised manuscript and indicated that the regression results should be interpreted with caution due to possible convergence issues.
Comment 4. Addressing these points will:
- Help clinicians interpret whether the MMPI-2 differences are actionable.
- Ensure the regression models are trustworthy and reproducible.
- Elevate the manuscript from “statistically interesting” to “clinically useful.”
Response 4. Thank you for these thoughtful comments, which we believe substantially enhance the transparency and interpretability of our study. Your suggestions contribute not only to improving the methodological rigor of the manuscript, but also to making its findings more clinically relevant and informative for practitioners.
Reviewer 2 Report
Comments and Suggestions for Authors
Authors made improvement to the study. Some figures are still unreadable and the font should be adjusted.
Reference list should be carefully checked.
Comments on the Quality of English LanguageEnglish language needs polishing.
Author Response
Comment 1. Authors made improvement to the study.
Response 1. Thank you for your positive feedback. We appreciate the reviewer’s acknowledgment of the improvements made to the study.
Comment 2. Some figures are still unreadable and the font should be adjusted.
Response 2. Thank you for this observation. We have updated the figures by replacing the transparent background with a white one to improve readability, especially in combination with the high-resolution format used. As for the font, we have maintained Palatino throughout, in accordance with the journal’s template guidelines. We hope these adjustments will address your concerns.
Comment 3. Reference list should be carefully checked.
Response 3. Thank you for pointing this out. We have carefully reviewed the reference list and corrected formatting inconsistencies, verified all citation details, and ensured that all references cited in the text are appropriately included and accurately reported. All entries have been formatted according to the style required by the journal and based on citation formats provided by the original source websites (e.g., publisher pages, journal platforms, or DOI registries).